# Network analysis of toxin production in *Clostridioides difficile* identifies key metabolic dependencies

**Deborah A. Powers[1], Matthew L. Jenior[2], Glynis L. Kolling[2], Jason A. Papin[1,2]***

**1** Biochemistry and Molecular Genetics, School of Medicine, University of Virginia, Charlottesville, Virginia, United States of America, **2** Biomedical Engineering, School of Engineering, University of Virginia, Charlottesville, Virginia, United States of America

* papin@virginia.edu

**Data Availability Statement:** Publicly available gene expression data from the GEO database was downloaded and used for this study; the GEO IDs are listed in S1 Table. The flux sampling, shadow pricing, and mMTA data are shared in S1, S2 and

## Abstract

*Clostridioides difficile* pathogenesis is mediated through its two toxin proteins, TcdA and TcdB, which induce intestinal epithelial cell death and inflammation. It is possible to alter *C. difficile* toxin production by changing various metabolite concentrations within the extracellular environment. However, it is unknown which intracellular metabolic pathways are involved and how they regulate toxin production. To investigate the response of intracellular metabolic pathways to diverse nutritional environments and toxin production states, we use previously published genome-scale metabolic models of *C. difficile* strains CD630 and CDR20291 (iCdG709 and iCdR703). We integrated publicly available transcriptomic data with the models using the RIPTiDe algorithm to create 16 unique contextualized *C. difficile* models representing a range of nutritional environments and toxin states. We used Random Forest with flux sampling and shadow pricing analyses to identify metabolic patterns correlated with toxin states and environment. Specifically, we found that arginine and ornithine uptake is particularly active in low toxin states. Additionally, uptake of arginine and ornithine is highly dependent on intracellular fatty acid and large polymer metabolite pools. We also applied the metabolic transformation algorithm (MTA) to identify model perturbations that shift metabolism from a high toxin state to a low toxin state. This analysis expands our understanding of toxin production in *C. difficile* and identifies metabolic dependencies that could be leveraged to mitigate disease severity.

## Author summary

*Clostridioides difficile* is the causative agent in approximately 73% of healthcare-acquired gastrointestinal infections, resulting in a significant healthcare burden. Its toxins are crucial to virulence and play a key role in establishing a nutritional niche for *C. difficile*. Highly virulent strains with high toxin production can lead to worse outcomes for patients with *C. difficile* infection (CDI), such as progression to pseudomembranous colitis, toxic megacolon, and in some cases, death. Improving our understanding of how these toxins are regulated through their environment and intracellular metabolism could allow us to

S3 Datasets respectively. The code used for this study is available on Github at https://github.com/dap5mb/cdToxinAnalysis.

**Funding:** The authors received funding from the following NIH grants: R01-AT010253 (MLJ, GLK, JAP) and T32-AI007046 (DAP). The funders had no role in study design, data collection and analysis, decision to publish, or preparation of the manuscript.

**Competing interests:** The authors have declared that no competing interests exist.

attenuate *C. difficile* virulence in infected patients. Therefore, we have compiled gene expression data of *C. difficile* grown in 16 different conditions to investigate how toxin production changes in response to the environment. We have integrated these data with a genome-scale metabolic model of *C. difficile*, allowing us to simulate the intracellular metabolism in high and low toxin producing states. Our network analysis of metabolism and toxin production predicts metabolic patterns in high and low toxin-producing states and provides insights into metabolic regulation of toxins. Additionally, our analysis highlights new proteins that could serve as anti-toxin targets.

## Introduction

*Clostridioides difficile* is the leading contributor to healthcare-acquired gastrointestinal (GI) infections, accounting for 73% of GI infections and costing an estimated $6.3 billion annually in the United States [1,2]. The primary risk factor for developing *C. difficile* infection (CDI) is broad-spectrum antibiotic usage which alters the structure and composition of the gut microbiota, allowing *C. difficile* to outgrow nonpathogenic competitors [3]. CDI is recurrent in 30% of cases and has a mortality rate of 9.3%, causing 29,000 deaths in the US annually [1,4].

CDI pathogenesis is primarily mediated by two large *Clostridia* toxins, TcdA and TcdB [5,6]. These toxins induce cytopathic and cytotoxic effects, including changes in epithelial cell morphology, cell cycle modulation, disruption of the colonic epithelial barrier, induction of apoptosis, and activation of an acute innate inflammatory response [6–10]. TcdA and TcdB induce a portion of this damage by glucosylating, and thereby inactivating, Rho GTPases in the host intestinal epithelial cells [7]. Rho GTPase inactivation disrupts the actin cytoskeleton and tight junctions of epithelial cells, resulting in a cytopathic phenotype of altered cell morphology and impaired epithelial barrier function [11]. A compromised colonic epithelial barrier can lead to increased intestinal permeability and fluid secretion which furthers intestinal inflammation and damage [5]. Together, these toxins are crucial to the establishment of CDI.

Regulation of *C. difficile* toxin synthesis is complex. TcdA and TcdB are encoded by genes on the pathogenicity locus (PaLoc) along with a transcriptional regulator, TcdR [12]. TcdR, in turn, is negatively controlled by metabolically sensitive global regulators such as CodY and CcpA which inhibit TcdR in the presence of intracellular branched chain amino acids (BCAAs) or fructose-biphosphate (FBP), respectively (Fig 1) [12,13]. In addition to these intracellular metabolites, the extracellular environment can also influence toxin production in *C. difficile*. Multiple defined media experiments with *C. difficile* demonstrate that certain carbohydrates, amino acids, and short chain fatty acids (SCFA) promote the increase or decrease of toxin production (Fig 1) [14–19]. Furthermore, *C. difficile* toxin production responses are dependent on the surrounding microbial community, as the microbial community shapes the nutritional environment of *C. difficile* [20]. Toxin response to environment may also be strain-dependent, further compounding the complexity of toxin production regulation [20]. While the exact relationship between the regulation of *C. difficile* toxin synthesis and extracellular environment is unclear, there is evidence that this process is linked to extra- and intracellular metabolism.

To investigate the metabolic states contributing to shifts in toxin production related to changes in the environment, we use previously established and curated genome-scale metabolic network reconstructions (GENREs) of *C. difficile* [21]. Metabolic modeling provides a unique systems approach to studying metabolism. Briefly, a GENRE describes the gene-protein-reaction associations for all of the metabolic genes of an organism which can then be

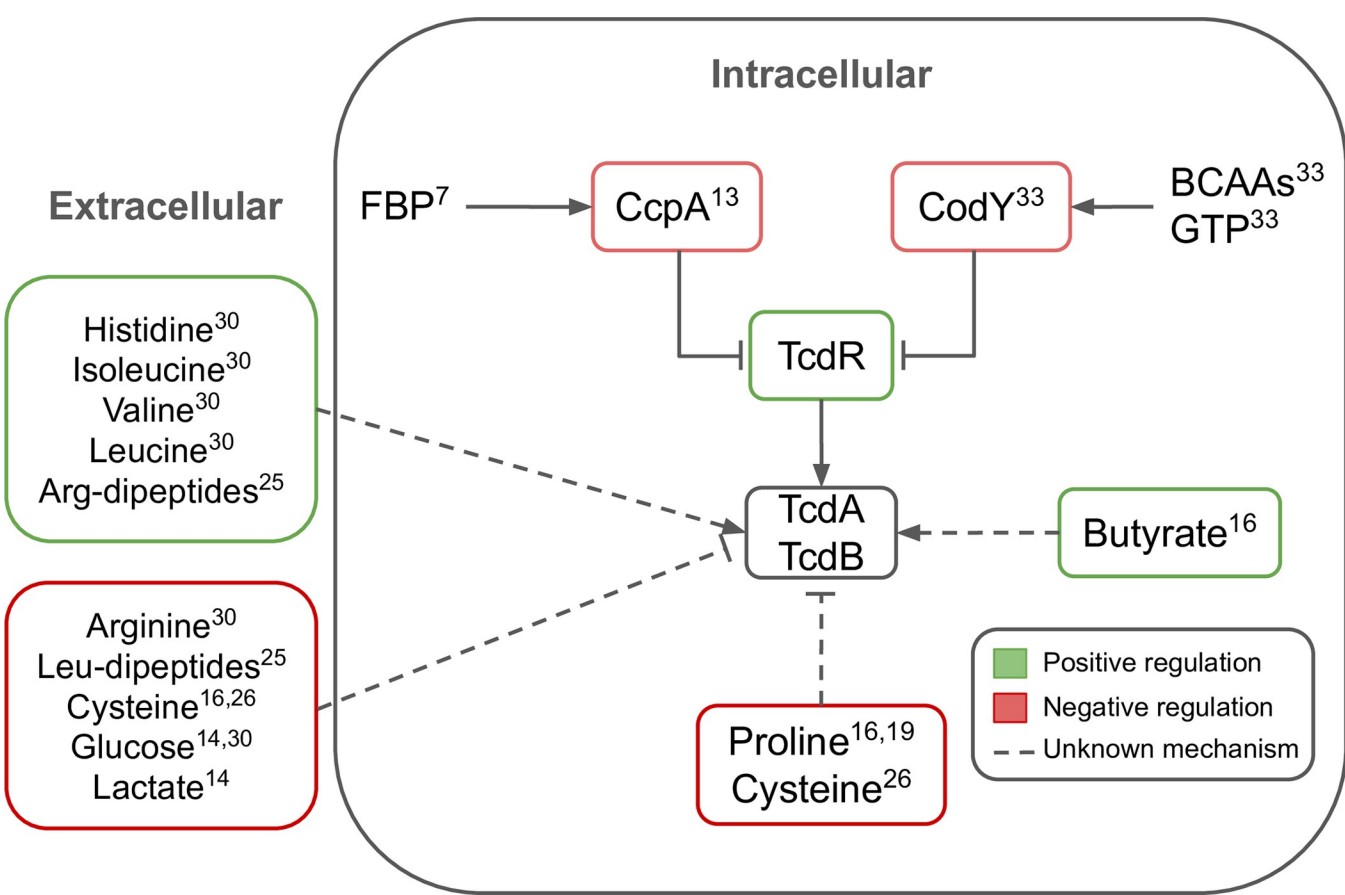

**Fig 1.** *C. difficile* **toxin production is regulated by multiple metabolic signals.** The transcription of *tcdA* and *tcdB* to synthesize toxins TcdA and TcdB is positively regulated by TcdR which is in turn negatively regulated by CcpA and CodY. Each of these components are regulated by multiple metabolic signals. Fructose bis-phosphate (FBP); branched-chain amino acids (BCAA); Guanosine triphosphate (GTP). Numbers correspond to references with evidence for the indicated regulatory mechanism.

simulated to predict the flux through the metabolic network [22]. We can represent the metabolic state of *C. difficile* in a specific context by integrating transcriptomic data from multiple studies (S1 Table) with the metabolic model via the RIPTiDe algorithm [23]. Using this approach, we created context-specific metabolic models of two *C. difficile* strains (CD630 and CDR20291) for 16 different environmental conditions with a range of toxin production states. We analyzed these models using a combination of machine learning and flux balance analysis (FBA) methods and found that arginine and ornithine metabolism is more active in models with low toxin production. Moreover, we found that arginine and ornithine metabolism may be influenced by intracellular fatty acid and large polymer pools. Finally, we applied a modified metabolic transformation algorithm (mMTA) to identify pivotal reactions for transitioning from a metabolic state associated with high toxin production to one associated with low toxin production [24]. Integrating gene expression data with a metabolic network simultaneously gives the gene expression data a metabolic context and the metabolic network biological relevance, optimizing our predictive power. Metabolic network analysis of microbial pathogens can catalyze biological discoveries which can then translate to new therapeutic leads.

## Results

### Contextualized metabolic models of *C. difficile*

Multiple studies have shown that toxin production in *C. difficile* can be altered by changing its nutritional environment; therefore, we wanted to investigate the intracellular metabolism of *C. difficile* grown in a variety of media conditions, inducing different toxin production states [13–16,18,20,25,26]. To do this, we used published GENREs of *C. difficile* strains CD630 and CDR20291 (iCdG709 and iCdR703, respectively) [21]. We compiled a set of publicly available RNA-sequencing data for each of these strains that considers a range of environmental conditions (S1 Table). For each condition, we classified the toxin states as low or high based on the median *tcdA* transcript reads per million (RPM) across all conditions (S1 Fig). Using the RIPTiDe algorithm [23], we integrated the transcript data with the appropriate strain model to generate a total of 16 contextualized models of *C. difficile* (Fig 2A).

On average, the RIPTiDe models for low toxin conditions include more genes than the high toxin RIPTiDe models but retain similar metabolite numbers (Fig 2A). The RIPTiDe models do share a core set of 483 reactions, which accounts for approximately 80% of the reactions in each model. There are strain differences, with CD630 models accounting for more genes, reactions, and metabolites compared to CDR20291 models (Fig 2A). Principle component analysis (PCA) of flux samples of these models demonstrates broad clustering by strain as well as by context (S2B–S2F Fig); complete flux sampling details can be found in the Materials and Methods. Overall, these models reflect metabolic differences by strain and toxin level with more metabolic genes and reactions included in both CD630 models and in low toxin models.

### Metabolic differences between strains and toxin states

To identify reactions important in distinguishing between metabolic states of low and high toxin conditions, we applied a Random Forest classifier to the flux sampling data (S3 Fig). Briefly, after randomly down-sampling the flux sampling data to 100 flux samples per RIPTiDe model, we used random stratified groups to split the flux sampling data, with a 75–25 train-test ratio (S3 Fig). The classifier had a mean accuracy of 97%; features were ranked by their Gini score to identify reactions important in distinguishing between toxin states (Materials and Methods). From the Random Forest analysis, we determined that arginine and ornithine reactions are more active in low toxin conditions (Fig 2B). Specifically, growth media supplemented with bile acid (deoxycholate and cholate) and calprotectin result in increased metabolic flux through arginine and ornithine transport reactions which move towards Stickland fermentation and NAD+ production via D-proline catabolism (Fig 2B and 2C). Other metabolic processes that are more active in low toxin conditions include reactions involved in carbohydrate metabolism and nucleotide metabolism (Fig 2B and 2C). However, for the majority of important reactions from Random Forest, the flux difference across all conditions was neither large nor significant. These results indicate that while metabolism can be predictive of toxin outcomes, it may be due to the cumulative effect of many tightly controlled reactions rather than large changes in flux through a few key reactions.

Because of the flux sampling results, we decided to investigate how sensitive growth and toxin production would be to disruptions in flux balance and intracellular metabolite concentrations. To approach this question, we conducted a shadow pricing analysis. Briefly, shadow pricing is the dual problem to flux balance analysis (FBA) in which shadow prices capture the sensitivity of an objective function (e.g., biomass) to changes in metabolites levels [27,28]. Thus, increases in levels of metabolites with negative shadow prices reduce flux through the objective function while increases in levels of metabolites with positive shadow prices increase

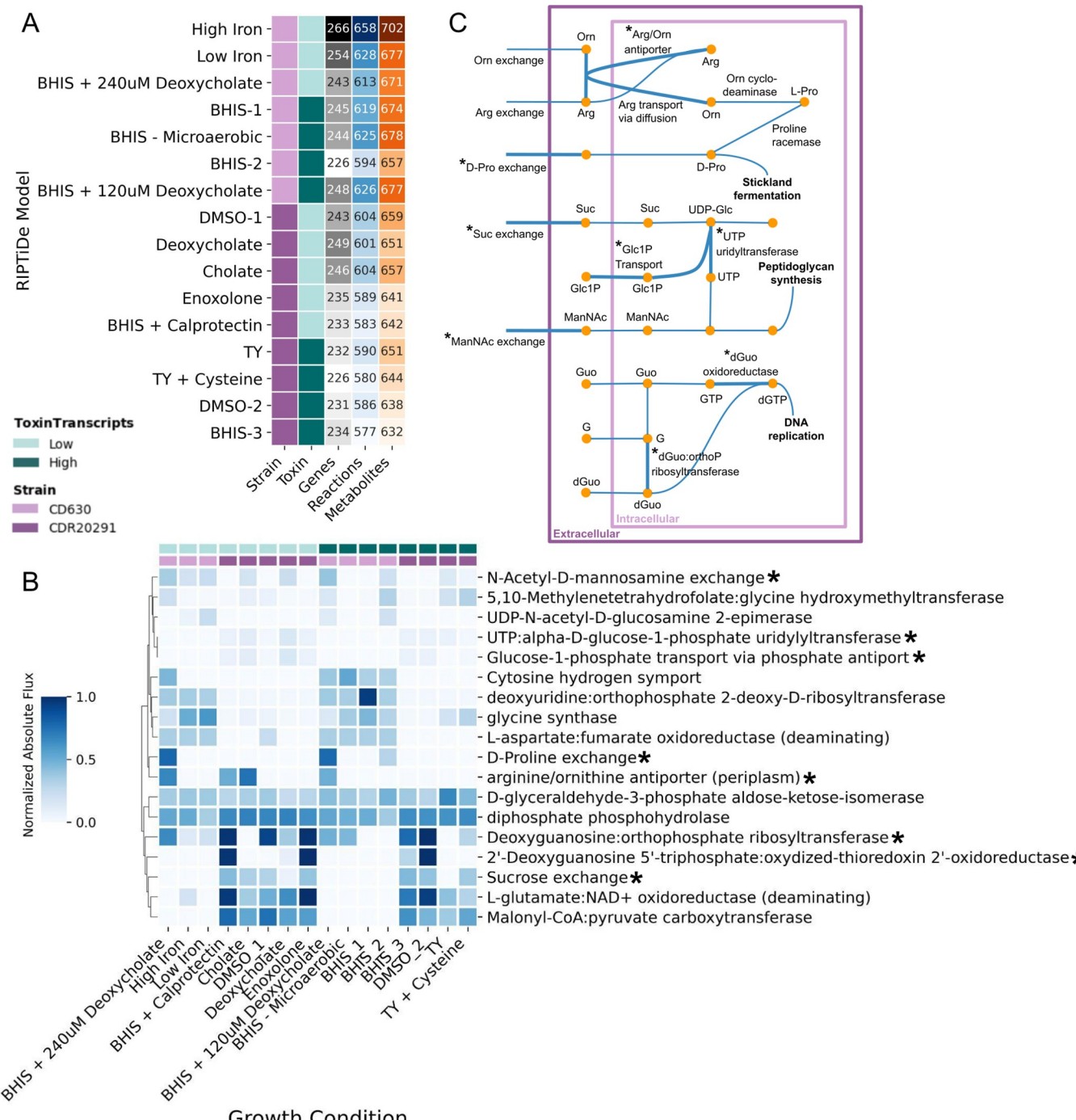

**Fig 2. Metabolic differences between toxin states are strain-specific.** (A) Summary table of the RIPTiDe contextualized models including the strain, toxin production level, and number of genes, reactions, and metabolites. (B) Normalized, absolute flux values for reactions indicated by Random Forest classifier as important for distinguishing between toxin levels. *C. difficile* strains 630 and R20291 are shown by light and dark purple respectively. Toxin transcript levels are shown by light (low) and dark (high) teal. Starred reactions are contextualized in panel (C). (C) Map of reactions in the metabolic model. Reactions identified by Random Forest analysis in panel (B) are starred. Arg: Arginine, Orn: Ornithine, Pro: Proline, Suc: Sucrose, UDP-Glc: UDP-Glucose, Glc1P: Glucose-1-phospate, ManNAc: N-acetyl-D-mannosamine, Guo: Guanosine, dGuo: Deoxyguanosine, G: Guanine.

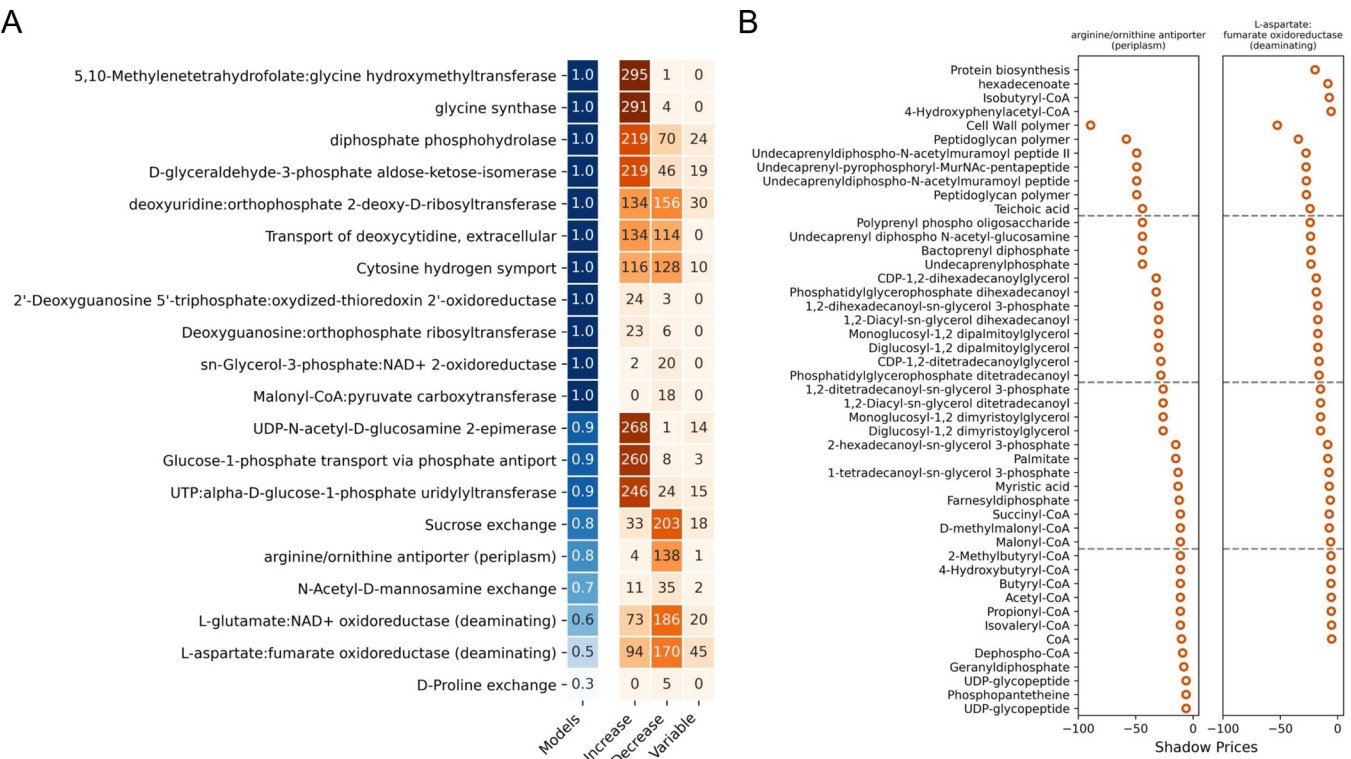

**Fig 3. Flux through arginine and ornithine reactions is sensitive to intracellular metabolite concentrations.** (A) Summary of the shadow pricing analysis with the top 20 reactions from the Random Forest classifier set as the objective function. The number in the "models" column (blue) corresponds to the fraction of contextualized models that were able to carry flux with the indicated reaction set as the objective function (OF). The values in the orange columns indicate the following: Increase: the number of metabolites for which an increased level results in increased flux through the OF (median shadow price > 0, range < 2); Decrease: the number of metabolites for which an increased level results in decreased flux through the OF (median shadow price < -0.1, range < 2); and Variable: the number of metabolites whose shadow price varied across RIPTiDe models (range > 2). For example, in the first row of panel (A), the OF was able to carry flux in all of the models, 294 metabolites increased flux through the OF in all of the models, 1 metabolite limited flux through the OF in all of the models, and 5 metabolites had different effects on flux through the OF across all of the models. (B) Shadow prices for limiting metabolites in arginine/ornithine and aspartate metabolism reactions. The metabolites categorized as sensitive in panel (A) for these OFs and with a shadow price < -5 are shown. Increasing negative values indicates increasing reaction flux sensitivity to the metabolite.

flux through the objective function. For this analysis we iteratively set the top 20 reactions from the Random Forest analysis as the objective function and solved the dual problem. Reactions whose flux is predominantly increased by increasing metabolite concentrations may indicate tightly regulated reactions where a consistent flux is important or perhaps that the reactions are not metabolically regulated. Conversely, reactions whose flux is decreased by many metabolites may be more responsive to the environment, allowing *C. difficile* to optimize its cellular function. Overall, we find that arginine and ornithine transport and L-aspartate: fumarate oxidoreductase are particularly sensitive to disruptions in metabolite concentrations with over 40 metabolites with a shadow price less than -5 (Fig 3). Both of these reactions are primarily sensitive to fatty acid and large polymer metabolite pools which have relatively high negative shadow prices (Fig 3B), indicating a possible source of intracellular metabolic regulation.

## Metabolic transformation between toxin states

From the flux sampling and shadow pricing analyses above, we described the metabolic state of *C. difficile* under 16 specific conditions and identified metabolic differences between low

and high toxin producing conditions. We next sought to investigate whether there were pivotal reaction knockouts that could transition the model from a metabolic state associated with a high toxin transcript level to a metabolic state associated with a low toxin transcript level. For this analysis we used a modified metabolic transformation algorithm (mMTA) which identifies key reactions to switch a cell from one metabolic state to another. To do this, MTA classifies reactions as changed or unchanged based on whether there was differential flux between a reference and target state; it then solves a mixed integer quadratic programming (MIQP) problem that maximizes change in flux in the direction of the target flux for the changed reaction set while minimizing flux changes for the unchanged reaction set [24]. We modified the original MTA for compatibility with COBRApy and to use updated modeling methods throughout its implementation. These changes and considerations are detailed in Materials and Methods. For our reference and target states we selected conditions that induce high and low toxin states respectively and that have transcriptomic data generated from the same study to reduce inter-study experimental variation. Therefore, we chose the iCdG709 RIPTiDe models contextualized with data from GSE165116, setting BHIS (high toxin) as the reference state and BHIS + 240 uM Deoxycholate (low toxin) as the target state (S1 Table, S1 Fig). Using mMTA, we classified 101 reactions as significantly different between the reference and target state. Out of this reaction set, 87 reactions were successfully changed in the desired flux direction in at least one reaction knockout simulation (S6C Fig). 396 out of 594 reaction knockouts resulted in a feasible solution; for each feasible reaction knockout, a transformation score (TS) was calculated (S6B Fig). The reaction knockouts with the top 50 TSs induced flux transformation in reactions related to energy metabolism, such as carbohydrate metabolism and Stickland fermentation (Fig 4). Of these transformed reactions, the Stickland fermentation reactions and most of the amino acid metabolism reactions are involved in isoleucine fermentation (Fig 4, S3 Dataset). Isoleucine is an important energy substrate that is metabolized via oxidative Stickland fermentation to form ATP [17]. This finding not only supports the importance of energy metabolism in toxin production, but also highlights the nutritional flexibility of *C. difficile* to acquire energy from available resources.

## Discussion

Research on *C. difficile* toxins has been extensive and wide-ranging, investigating biochemistry and structure, mechanisms of action, host and microbiome interactions, nutritional environments, genetic and metabolic regulation, and more [6,10–12,20,29]. Of these, the link between metabolism and regulation is particularly salient because of its therapeutic potential for virulence attenuation. Many *in vitro* studies have shown that *C. difficile* toxin production can be manipulated through its environment [13–16,18,20,30,31]. Additionally, multiple mechanisms of intracellular toxin regulation have been shown, such as cyclic di-GMP, carbon catabolite repression via *ccpA*, and nutritional limitation via *codY* [13,32,33]. However, while all of these studies have elucidated various facets of *C. difficile* toxin regulation under specific conditions, our understanding of the relationship between metabolism and toxin regulation remains fragmented. To address this, we conducted a systems analysis in which we interrogated an array of environmental conditions and toxin transcript levels with the goal of defining the metabolic state of *C. difficile* in these conditions.

The role of arginine in *C. difficile* toxin production is contradictory. Early investigation of *C. difficile* grown in minimal defined media supplemented with arginine found that increasing arginine concentrations resulted in decreased toxin production and enhanced growth [30,31]. Conversely, a study using phenotype microarrays found that arginine, and arginine dipeptides in particular, induce toxin production [25]. When we simulated growth under a subset of PM

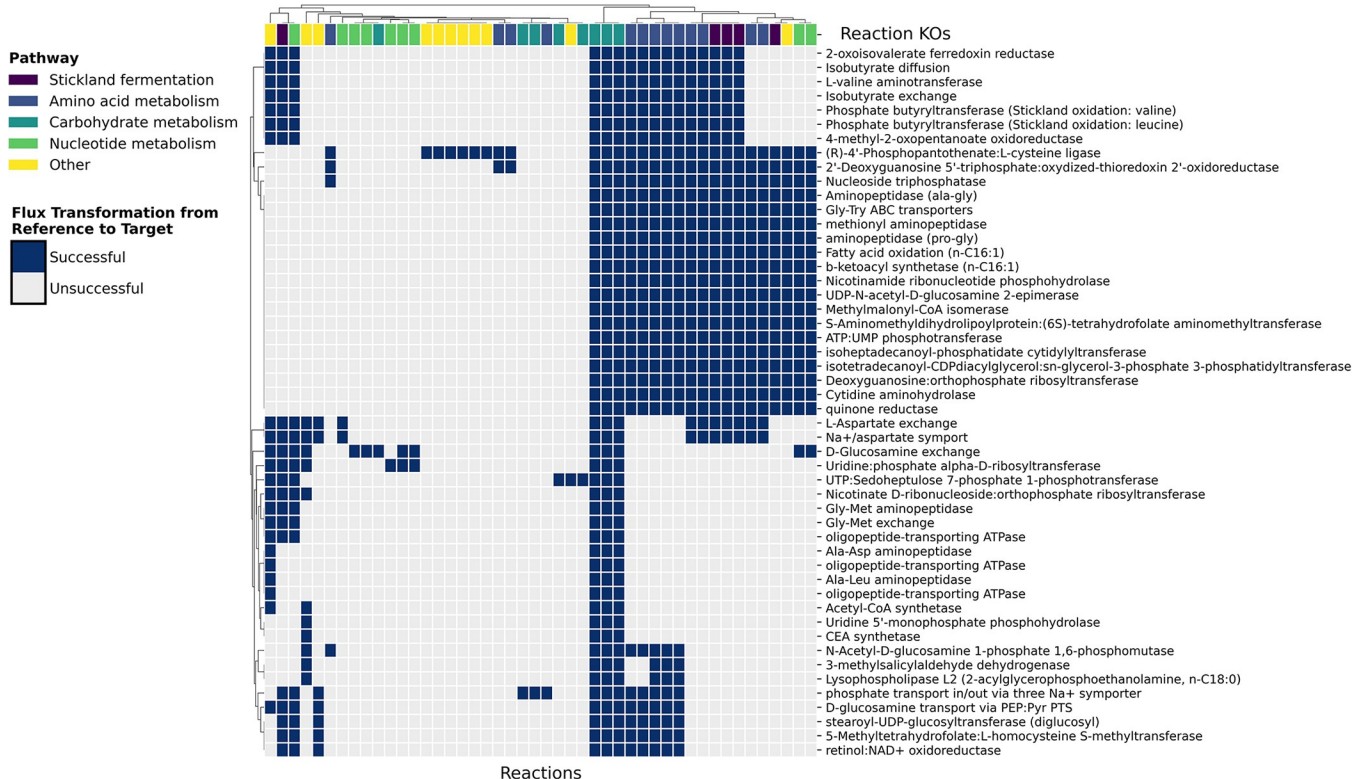

**Fig 4. Modified MTA identifies key reaction knockouts and pathways for transformation from a high to low toxin state.** The mMTA algorithm runs a reaction KO simulation to optimize changes in reaction flux that transform the model from the reference metabolic state (high toxin) to the target metabolic state (low toxin). The reaction knockouts with the highest transformation scores are shown on the y-axis. The reactions whose flux changed under these KO conditions are shown on the x-axis. Successfully changed reactions are defined as those whose flux changed from the reference in the desired direction by a minimum threshold of significance (successful: dark blue; unsuccessful: light grey). The metabolic pathways for these reactions are shown beneath the clustering dendrogram at the top.

conditions, we did not find any correlations between metabolic flux and the PM toxin data (S8 Fig). The only model constraints in this simulation were growth limits rather than intracellular limits as for the RIPTiDe models; these constraints, combined with the differences in which strain was profiled with the PM experiments and which strains were modeled with the analysis presented here, may have contributed to this lack of correlation. Cecal metabolomics of mice infected with wild-type (inflammatory) and toxin-deficient (non-inflammatory) *C. difficile* strains show that metabolic pathways for arginine and ornithine in the gut microbial community are more active in the non-inflammatory state [34]. This same study also linked ornithine metabolism in *C. difficile* with reduced inflammation during CDI. In our analysis, an arginine/ornithine transport reaction was identified as an important differentiator between toxin states by Random Forest and is particularly active in low toxin states across strains (Fig 2B). Shadow price analysis of these reactions found it was highly sensitive to fatty acids or large polymers primarily involved in cell wall synthesis (Fig 3B). Fatty acids have been shown to regulate arginine and ornithine metabolism in other organisms [35–37]. This potential regulatory interaction between fatty acids and arginine in *C. difficile* could explain the contradictory results for the impact of arginine and ornithine metabolism on toxin production.

Toxin production in *C. difficile* is clearly linked to its nutritional status. We used Random Forest to attempt to identify any underlying metabolic patterns between toxin states but to understand better the intracellular metabolic switches necessary for transitioning from a high

to low toxin production state we used mMTA. The results from mMTA give us two types of information. First, the metabolic reactions whose activity is important for this transition and second, the metabolic reactions that can be modulated to induce these metabolic changes. We found that within the metabolic network model corresponding to *C. difficile* grown in BHIS with high toxin production, flux through 86% of the reactions identified as important for transitioning to a low toxin state could be modulated by at least one reaction knockout. With the Random Forest analysis we performed, we found that the 20 reactions with the highest Gini scores are heavily involved in energy metabolism (e.g., arginine/ornithine, glycine, glutamate, aspartate, sucrose, glucose, and N-acetyl-D-mannosamine) as well as nucleotide metabolism (e.g., guanosine and uridine). Similar patterns of important metabolic pathways are replicated in the mMTA results; reactions whose flux can be changed to mimic a low toxin state fall into carbohydrate, amino acid, and nucleotide metabolism categories (Fig 4). The mMTA results also predict that these reactions can be metabolically modulated via knockouts of specific reactions (as indicated on the y-axis in Fig 4). The reaction knockouts with the highest transformation scores were frequently key reactions in alternative energy-generating pathways such as carbon metabolism and Stickland fermentation of leucine and valine (Fig 4). When these reactions were knocked out, Stickland fermentation of isoleucine increased (S3 Dataset).

Isoleucine is another metabolite whose role in toxin production in *C. difficile* is unclear. Isoleucine activates CodY which represses toxin production (Fig 1) [38]; it is reasonable to hypothesize that conditions supplemented with isoleucine would have low toxin production. However, defined media experiments show the opposite effect. *C. difficile* (VPI 10463) grown in a minimal media supplemented with isoleucine demonstrated increased TcdA production [31]. Another study growing *C. difficile* (ATCC 9689) in phenotype microarrays found that isoleucine induced middle-level toxin production [25]. It may be that preferential fermentation of isoleucine throughout the exponential growth phase depletes stores of bioavailable isoleucine for CodY activation in stationary phase, resulting in CodY deactivation and increased toxin production. While our mMTA results show that isoleucine fermentation was maximized in the target low toxin state, perhaps the driving differential feature between states is carbohydrate metabolism (S7 Fig). In the reference state, increased glucose metabolism is likely driving the high toxin transcript levels [14]; in the target state, suppression of glucose metabolism is accompanied by an increase in isoleucine fermentation to maintain energy generation. In the short-term, this increased isoleucine fermentation likely results in increased uptake and availability of isoleucine which is sufficient for CodY activation and suppression of toxin transcription.

We used *C. difficile* GENREs and publicly available transcriptomic data for our analyses, applying RIPTiDe for data integration. RIPTiDe uses genetic evidence from the transcriptomic data as weights indicating the likelihood that a reaction will carry flux and to what extent [23]. It additionally prunes reactions for which there is no evidence or that do not pass a minimum flux threshold. This approach creates a contextualized model for a specific environmental and metabolic state. However, it is possible that this method may have inadvertently limited the results of our mMTA analysis. The flux bounds of reactions in a RIPTiDe model are set based on genetic evidence and are often quite restrictive because the goal of RIPTiDe is to describe the metabolic state of an organism in a specific condition. Therefore, any significant departure from the reference state flux necessary for a reaction to achieve a target state may not even be feasible due to preset flux bounds. However, we were able to successfully transform 86% of the reactions targeted for change, and therefore do not consider any potential limitation from application of RIPTiDe as having a significant impact on our mMTA results.

We additionally used RIPTiDe to sample the flux distributions for each model which we then used for our Random Forest analysis. The flux samples for each condition are highly

correlated in part due to the RIPTiDe restrictions discussed above; this characteristic could lead to overfitting in a Random Forest model because the model would be able to learn what condition a sample is from and then use this information to infer the toxin level. To prevent this, we used a random stratified group sampling approach for splitting the data into train-test sets for Random Forest (Methods). This approach ensures first that there is an equal (or near-equal) ratio of low and high toxin conditions in the train and test sets and second that all the flux samples from a single RIPTiDe model are used in either the train or test set. The classifier had a mean accuracy of 95% which suggests model over-fitting despite the feature selection and sampling approaches we took. A closer look at the features with the highest Gini scores shows that there is a greater difference in flux between strains than in flux between toxin states (Fig 2). While the classifier is not learning what condition a sample is from, it may be learning what strain it is and then predicting the toxin state based on strain-specific criteria. This result highlights the importance in accounting for strain differences when interrogating toxin production in *C. difficile*.

The reactions from the Random Forest Classifier with the highest Gini scores were analyzed in a shadow price analysis. The results of this analysis showed that arginine and ornithine transport as well as an aspartate fumurate oxidoreductase were highly sensitive to fatty acids and large polymers. However, two metabolites that also commonly occurred as limiting (shadow price < -5) were "Protein biosynthesis" and "Cell Wall Polymer" (Figs 3 and S5). These metabolites are not true biological metabolites but rather *in silico* substitutions for high-level cellular processes just upstream of the biomass reaction within the model. While the biomass reaction is not set as the objective function in the shadow pricing analysis, there may be underlying biases that drive flux towards biomass production. Regulation via these two "metabolites" may indicate an intracellular sensing mechanism or be used as proxy for cell status but it is also possible that they are merely modeling artefacts.

In conclusion, we performed a systems analysis of *C. difficile* metabolism under different growth conditions, paired with the associated toxin transcript level to define the relationship between metabolism and toxin production. These toxins are essential in establishing a nutritional niche for *C. difficile* and can cause extensive damage in the host colon. CDI is most effectively resolved through fecal microbiota transplants (FMTs) [39]; however, FMTs are typically only prescribed for patients with severe or recurrent cases of CDI [39]. Using microbial engineering to design probiotic communities that can be offered as a non-invasive CDI therapeutic is a major advancement already underway within the field [40,41]. An important step in designing these therapeutic communities is identifying reactions or pathways associated with high and low toxin production and understanding how those reactions change as a function of the environment, resulting in specific toxin-associated phenotypes. Future research investigating questions include accounting for the relationship between regulatory networks and metabolism in *C. difficile* toxin production as we know that toxin responses are in part the effect of global regulators. Additionally, modeling the effect of toxin activity once it is released from *C. difficile* could also help guide selection of members of a microbial community. Research in these areas will provide foundational understanding of *C. difficile* biology that will enable intentional and specific therapeutic community design.

## Materials and methods

### Processing the RNA-sequencing data

We compiled transcriptome count matrices from seven publicly available RNA-sequencing studies of *C. difficile* covering two strains (630 and R20291) and 16 unique conditions

(S1 Table). Raw count matrices were normalized using the reads per million (RPM) formula

$$\frac{\text{Number of reads mapped to gene x } 10^6}{\text{Total number of mapped reads}}.$$

RPM normalized toxin gene transcripts from all studies were grouped by condition, averaged, and binned into high and low categories by *tcdA* medians (low < median < high) (S1 Fig). We used medians of *tcdA* rather than *tcdB* because *tcdB* had low to no expression across all conditions with the exception of the tryptone yeast conditions (TY and TY + Cysteine).

## Genome-scale metabolic models

Previously published *C. difficile* genome-scale metabolic network reconstructions (GENREs) of strains CD630 and CDR20291 (referred to as iCdG709 and iCdR703, respectively) were used for all of the modeling analyses [21]. We created a total of 16 contextualized metabolic models of *C. difficile* using RIPTiDe and the processed transcriptomic read matrices from publicly available RNA-Seq studies (S1 Table) [23].

## Random Forest analysis of flux samples

For each RIPTiDe model, we optimized for biomass production and then sampled (n = 500) flux distributions of the entire feasible steady-state solution space using RIPTiDe. We then randomly down sampled to 100 flux samples per condition and performed a principal component analysis (PCA) of sampled flux distributions for all the models.

To identify the most important reactions in differentiating between toxin production (low, high) we ran a Random Forest classifier with 500 trees on the down-sampled flux sampling data (100 flux samples per RIPTiDe model). We reduced the feature space by selecting features with a near-zero variance (NZV) > 0.005 and an absolute Pearson's correlation coefficient < 0.8. We used random stratified group K-fold (k = 5) cross validation to check the classifier (S3 Fig). Using a stratified group k-fold to split the data into train and test sets ensures first that there is an equal (or near-equal) ratio of low and high toxin conditions in the train and test sets and second that all the flux samples from a single RIPTiDe model are used in either the train or test set. We used this approach to prevent the classifier from learning which RIPTiDe model the flux sample came from and substituting that information to infer toxin level; while there are many flux samples per RIPTiDe model, these flux samples tend to be highly correlated. For model predictions, the data was split using a random stratified group split as described for the cross validation. The classifier was then trained on 75% of the data and tested on the remaining 25%.

Following classifier testing, we ranked the features by their Gini score and selected the top 20 most important features for model predictions. For these 20 reactions, we calculated the median flux value for each condition, normalized, and visualized using a heatmap (Fig 2B). We tracked the flow of flux through these reactions to create a human readable metabolic map (Fig 2A) as well as an Escher metabolic map [42] with the GENRE IDs for the reactions and metabolites (S4 Fig). To investigate metabolism of *C. difficile* that literature indicates can impact toxin production, we compiled lists of identifiers for reactions within a specific metabolic pathway. We did this analysis for three metabolic processes: Stickland fermentation, ATP production, and redox reactions. We filtered and processed the flux sampling data in the same way as for the Random Forest results.

## Shadow pricing analysis

The 20 reactions with the highest Gini scores from the Random Forest analysis were iteratively set as the objective function for each RIPTiDe model. This model was then optimized and the corresponding shadow price for the FBA solution was saved. For each metabolite and each objective function, we calculated the median shadow price and range across all RIPTiDe models. We summarized the shadow pricing results for each objective function (OF) across all RIP-TiDe models using the following metrics: fraction of RIPTiDe models able to carry flux for that OF, total number of metabolites that increase flux through the OF (median shadow price < -0.1, range < 2), total number of metabolites that decrease flux through the OF (median shadow price > 0, range < 2), total number of metabolites whose shadow price varied across RIPTiDe models (range > 2). We then plotted the metabolites with a median shadow price < -5 and a range < 2 for the 9 objective functions that had metabolites in this category (Figs 3B and S5).

## Metabolic transformation algorithm (MTA)

The goal of MTA is to identify perturbations that transform a metabolic network from a reference state (e.g., diseased) to a target state (e.g., healthy) [24]. In our case, we used the MTA to find reaction knock-outs that lead to transformation of high toxin states to low toxin states in *C. difficile*. The generic MTA is comprised of four distinct steps which are briefly: (1) calculate a flux solution for the reference state, $v_{ref}$, (2) identify reactions that are changed in the forward ($R_F$) or backward ($R_B$) direction and unchanged ($R_S$) between the reference and target state, (3) solve the MIQP optimization problem formulated to minimize change in $R_S$ and maximize change in $R_F$ and $R_B$, and (4) calculate a transformation score (TS) to quantify the success of each reaction knockout in transforming the reference state to the target state. To successfully apply MTA to our problem, we made changes to the original formulation at each step, resulting in a modified MTA (mMTA, described below) compatible with COBRApy tools.

*Step 1*: We created two contextualized metabolic models (reference and target) using RIP-TiDe with gene expression data and then generated 500 flux samples using RIPTiDe [23]. Because the mean (or median) of the flux samples is not a mass-balanced solution, setting it as $v_{ref}$ can lead to infeasible MIQP solutions downstream. Therefore, we used a Bray-Curtis non-parametric multidimensional scaling (NMDS) to reduce the flux samples to a two-dimensional space, then calculated the centroid of the flux sampling distribution, and finally calculated the point closest to the centroid and set this flux sample as $v_{ref}$ (S6A Fig).

*Step 2*: We determine significantly changed and unchanged reactions by using a Mann-Whitney U test with a Bonferroni multiple tests correction. We categorize all reactions into three sets: statistically insignificant reactions ($R_S$), and statistically significant reactions which must change in the forward ($R_F$) or backward ($R_B$) direction in order to match the target state. The threshold for statistical significance is an adjusted p-value < 0.05.

*Step 3*: The goal of the MIQP problem is to minimize changes in flux for reactions in $R_S$ and maximize changes in flux for reactions in $R_F$ and $R_B$ in the intended direction. We implemented the MIQP formulation as it was set out for the generic MTA:

$$\min_{v,y}\left((1 - \alpha) \sum_{i \in R_S} (v_i^{ref} - v_i)^2 + \frac{\alpha}{2} \sum_{i \in R_F} y_i + \frac{\alpha}{2} \sum_{i \in R_B} y_i\right)$$

$$s.t.$$

$$S \cdot v = 0$$

$$v_{min} \leq v \leq v_{max}$$

$$v_j = 0$$

$$v_i - y_i^F(v_i^{ref} + \varepsilon_i) - y_i v_i^{min} \geq 0, i \in R_F$$

$$y_i^F + y_i = 1, i \in R_F$$

$$v_i - y_i^B(v_i^{ref} - \varepsilon_i) - y_i v_i^{max} \geq 0, i \in R_B$$

$$y_i^B + y_i = 1, i \in R_B$$

$$y_i, y_i^F, y_i^B \in \{0, 1\}$$

Mass balance constraints, thermodynamic constraints, and the reaction knockout perturbation are enforced in equations 2, 3, and 4 respectively. The demands for changed reactions are represented in equations 5–8, such that the Boolean variables $y_i, y_i^F, y_i^B$ indicate whether a forward reaction ($R_F$) either increases by more than $\varepsilon$ with respect to $v^{ref}$ or maintains a preset flux minimum and whether a backward reaction ($R_B$) either decreases by more than $\varepsilon$ with respect to $v^{ref}$ or maintains a preset flux maximum. $\varepsilon$ is the vector of thresholds used to determine if flux changes are statistically significant ($p < 0.05$) and was calculated using a one-sided T-test with a 95% confidence interval, such that $\varepsilon = t\frac{S}{\sqrt{n}}$ where t is the critical value, S is the standard deviation, and n is the number of flux samples. We set $\alpha = 0.66$ as in the original formulation.

*Step 4*: We categorize forward and backward reactions as successful if $v_{RF} > (v_{RF}^{ref} + \varepsilon)$ or if $v_{RB} < (v_{RB}^{ref} - \varepsilon)$, respectively. Next, to quantify how well each reaction knock-out transformed the reference state to the target state, we calculated the transformation score (TS) as formulated for the generic MTA:

$$\frac{\sum_{i \in R_{Success}} abs[(v_i^{ref} - v_i^{res}] - \sum_{i \in R_{unsuccess}} abs[(v_i^{ref} - v_i^{res})]}{\sum_{i \in R_S} abs[(v_i^{ref} - v_i^{res})]}$$

Ranking reaction knockouts by the TS provides a helpful metric for evaluating the flux solution from each knockout while taking the MIQP objective value would result in reaction knockouts with varying degrees of transformational success being set as equivalent (S6B Fig).

## Phenotype Microarray (PM) data integration and analysis

We used public data from a PM study that measured toxin production of *C. difficile* type strain ATCC 9689 when grown in each condition [25]. The toxin concentrations from this study were calculated by comparing the amount of dye reduction in cell cytotoxicity assays to a standard curve of toxin concentrations [25]. The authors defined toxins as low (<42 ng/uL), mid (42–420 ng/uL), or high (>420 ng/uL) and we used the same categories in this analysis. The dataset from this PM study included 652 unique growth conditions. The GENREs iCdG709 and iCdR703 contain 171 unique extracellular metabolites, 65 of which overlapped with metabolites from the PM dataset (S8A Fig). We constrained the GENREs to minimal media conditions and iteratively added one of the 65 overlapping metabolites and simulated flux while optimizing for biomass. We normalized the flux sampling data using min-max

normalization and then removed reactions with variance < 0.05. This step trimmed the flux data from 1323 reactions to 67 reactions. Next, we calculated the Pearson's correlation between each reaction flux vector and the PM toxin data. None of the reactions were correlated with toxin production. Finally, we visualized the absolute flux data for each of the 65 simulated PM conditions (S8C Fig).

## Supporting information

**S1 Table. Public RNA-sequencing datasets.** BHIS(G): Brain-Heart Infusion Supplemented (Glucose), Cd: *C. difficile*, CDMM: *C. difficile* Minimal Media, DCA: Deoxycholate, DMSO: Dimethyl Sulfoxide, GEO ID: Gene Expression Omnibus Identifier, TY: Tryptone Yeast. Alternate identifiers for RIPTiDe models with similar growth conditions are indicated in parentheses in the Growth Condition column when applicable; these identifiers are used for all analyses.
(DOCX)

**S1 Fig. Toxin transcript counts across conditions.** Toxin transcript counts quantified as reads per million (RPM) are shown for all conditions included in the study (see S1 Table for more details). Conditions were binned based on median *tcdA* transcript levels across all conditions and labeled as low (< median) or high (> median).
(TIFF)

**S2 Fig. PCA of flux sampling of RIPTiDe-contextualized iCdG709 and iCdR703 models.** (A) Summary table of the RIPTiDe-contextualized models including the strain, toxin production level, and number of genes, reactions, and metabolites. (B) The iCdG709 (CD630, light purple) and iCdR703 (CDR20291, dark purple) *C. difficile* models were contextualized with transcriptomic data (S1 Table) and flux distributions were sampled (n = 500) using RIPTiDe. The flux sampling for each model was randomly down-sampled to 100 flux samples and PCA was performed for all the models together (B) and by strain (C–F).
(TIFF)

**S3 Fig. Random Forest validation metrics.** (A) Visualization of the random stratified group k-fold splits used for cross validation of the Random Forest classifier. (B-C) K-fold cross validation (k = 5) of the Random Forest classifier testing ROC (B) and accuracy (C), with an average accuracy of 95% in cross validation. (D) Confusion matrix for model predictions with train and test sets selected in a 75–25 ratio using random stratified group splits. The model trained on this set had a 97% accuracy. (E) The top 20 features for model predictions by Gini score.
(TIFF)

**S4 Fig. Escher metabolic maps.** Metabolic context for reactions from the Random Forest analysis labeled with the reaction and model IDs from the GENREs iCdG709 and iCdR703.
(TIFF)

**S5 Fig. Shadow prices of metabolites that decrease flux through reactions from Random Forest.** For each objective function (OF) listed in Fig 3A, the metabolites categorized as decreasing and with a shadow price < -5 are shown.
(TIFF)

**S6 Fig. MTA calculations for centroids and transformation scores (TS).** (A) Bray-Curtis NMDS of flux sampling results for iCdG709 contextualized for BHIS + DCA 240 uM (target, low toxin, light teal) and BHIS (reference, high toxin, dark teal) was used to calculate the

centroids (red) and the flux sample closest to the centroid (orange) for each model. (B) The MIQP objective value verses the TS demonstrates the utility of the TS in ranking flux solutions with a similar objective-value based on success of the flux solution in transforming reactions to the target state. (C) Successfully changed reactions for each reaction knockout. Successful (dark blue), unsuccessful (light blue).
(TIFF)

**S7 Fig. Comparison of metabolic flux through reactions in the Reference and Target state.**
(TIFF)

**S8 Fig. Phenotype microarray (PM) simulation and analysis.** (A) Venn diagram showing the overlap of unique metabolites from the PM dataset and the extracellular metabolites from the GENREs. (B) The toxin concentration distribution for the 65 overlapping growth conditions from panel (A). (C) Simulated reaction flux through each *in silico* PM condition (n = 65). The flux data was min-max normalized and reactions with flux variance across all conditions $< 0.05$ were removed and the absolute flux value of the remaining reactions was visualized. The PM growth conditions are sorted by their toxin category. Toxin categories were defined as low ($<42$ ng/uL), mid (42–420 ng/uL), and high ($>420$ ng/uL) as in Lei, XH and Bochner, BR (2013).
(TIFF)

**S1 Dataset. RIPTiDe model flux sampling data.** Down-sampled flux data (n = 100 samples per RIPTiDe model), with the first three columns set as sample descriptors (condition (RIPTiDe model), strain, and toxin category).
(CSV)

**S2 Dataset. Shadow pricing data.** Metabolite shadow prices with the first five columns set as simulation descriptors: condition (RIPTiDe model), strain (CD630 or CDR20291), toxin (low or high), OF (reaction ID for objective function), and OF_name (name of OF).
(CSV)

**S3 Dataset. MTA knockout flux data.** Flux data for each reaction knockout (columns) with the first two columns showing the flux data for the Target and Control (Reference).
(CSV)

## Author Contributions

**Conceptualization:** Deborah A. Powers, Jason A. Papin.

**Data curation:** Deborah A. Powers.

**Formal analysis:** Deborah A. Powers.

**Investigation:** Deborah A. Powers.

**Methodology:** Deborah A. Powers, Matthew L. Jenior.

**Project administration:** Jason A. Papin.

**Resources:** Jason A. Papin.

**Software:** Deborah A. Powers, Matthew L. Jenior.

**Supervision:** Glynis L. Kolling, Jason A. Papin.

**Visualization:** Deborah A. Powers.

**Writing – original draft:** Deborah A. Powers.

**Writing – review & editing:** Matthew L. Jenior, Glynis L. Kolling, Jason A. Papin.

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
