## [Decision Letter · Decision Letter 0]

30 Jan 2023

Dear Professor Papin,

Thank you very much for submitting your manuscript "Network analysis of toxin production in Clostridioides difficile identifies key metabolic dependencies" for consideration at PLOS Computational Biology.

As with all papers reviewed by the journal, your manuscript was reviewed by members of the editorial board and by several independent reviewers. In light of the reviews (below this email), we would like to invite the resubmission of a significantly-revised version that takes into account the reviewers' comments.

We especially recommend you to comprehensively address the following issues: a) ML overfitting, b) empty github repository, and c) integration of phenotypic data.

We cannot make any decision about publication until we have seen the revised manuscript and your response to the reviewers' comments. Your revised manuscript is also likely to be sent to reviewers for further evaluation.

Sincerely,

Kiran Patil

Section Editor

PLOS Computational Biology

We especially recommend you to comprehensively address the following issues: a) ML overfitting, b) empty github repository, and c) integration of phenotypic data.

Reviewer's Responses to Questions

**Comments to the Authors:**

Reviewer #1: Summary of Research:

The authors present a computational analysis of metabolic regulation of toxin (TcdA and TcdB) production in C. difficile. They integrate public transcriptomics data to develop 16 context specific metabolic models, using the RIPTiDe algorithm and C. difficile metabolic models. These base models and algorithm were previously developed by several of the same authors. They analyze the models using a combination of flux sampling, machine learning (random forest), and shadow prices to identify patterns associated with toxin production. They also implement the metabolic transformation algorithm to identify reaction knockouts that drive metabolic flux towards a low-toxin state. The discussion of the results focuses on arginine and ornithine transporters and isoleucine. Overall, the scope and motivation of the work is interesting. The conceptual model is set up well. The analysis and interpretation of the results should be explained more clearly. Specific major and minor comments are provided below.

Major Comments:

1. I am concerned with overfitting in the machine learning results. The total number of samples is quite small (16). I understand that there are additional samples created through the flux sampling process, but these will likely be highly correlated. The model can likely learn, from these correlated samples, which condition a flux sample comes from and use that information to infer the toxin level. Therefore, the degree to which the results correspond to microbial physiology in toxin production vs condition specific physiology is not clear. This should be addressed in the discussion.

2. To further address overfitting to condition in the results, I would also suggest that the authors implement a cross-validation scheme where flux samples from the same condition are not mixed across the train and validate sets. To do this you can randomly select a set of conditions (of the 16) to assign to the training set (repeating random selections for statistics). The random selection could be stratified to ensure there are always some high and some low toxin samples. Then use all the flux samples from the randomly selected conditions in the training set and the remaining samples in the validate sets. Cross-validating in this way will give insight into whether the ML model can generalize across conditions and should be presented alongside the current results.

3. Another suggestion for the machine learning analysis is to reduce feature correlations. ML algorithms, and in particular feature importance calculations, are typically more stable when the input features are not correlated. Before implementing the ML the authors can clean up the input flux features in two ways. First, reduce the number of features by getting rid of any flux that has variability across all samples below some specified threshold (these features contain little to no information). Second, cluster features with covariance across all samples above some threshold into one feature (one of these fluxes can be used as a representative feature for the machine learning). There are often linear pathways of reactions with highly correlated flux across samples that can be reduced in this way. Reducing the feature space could make the results more stable and easier to interpret. If any feature engineering of this sort has already been implemented, the authors should include the details in the methods section.

4. The authors could further discuss the relationship between the ML analysis and the MTA analysis. What are the differences in the types of results you expect to find with these two approaches? How do the results from the two approaches complement or support each other?

Minor Comments:

1. Line 11. Consider changing “its toxins” to “two toxic proteins”. Defining TcdA,B more specifically as proteins (as opposed to metabolites) will help non-expert readers quickly understand the context of the modeling work.

2. Line 51-54. This is not a comment for this paper, but I was curious if the glucosylating activity of TcdA,B is a metabolic process that could be modeled. Is that included in the genome-scale metabolic models? Just a thought on another possibly interesting line of inquiry.

3. Line 81-83. It would be nice to include in the introduction a bit more description of the transcriptomic data that was used. Is this all from one study, multiple studies? Is there any motivation behind the different conditions that were included? Maybe just a reference to Supplemental Table 1 would suffice.

4. Line 98-99. Is there any reason for choosing tcdA over tcdB? Are tcdA and tcdB correlated across conditions?

5. Line 107-108. Please provide additional description of the flux sampling here or point to this description in the methods. Was this sampling of the entire feasible steady-state space, was biomass production optimized and used as a constraint?

6. Paragraph 111-121. The machine learning approach should be described in more detail. How many sampled flux distributions were used for each condition? What was the train/validate/test split procedure? What was the accuracy of the ML model on the test (or validation) set relative to a null distribution? How was feature importance extracted from the random forest model? This information is important for interpreting the results so it should be presented to some degree in the results section.

7. Line 122. Figure 2. It may be nice to highlight the arginine and ornithine transport related reactions. The figure in part B does a nice job of explaining the context of the reactions from the ML importance results in part A. Maybe highlight the relevant reactions in A with bold or a different colored font and link them to the matching reaction in part B with a superscript.

8. Paragraph 130-141. It would be nice to have more discussion of why certain reactions have many sensitive metabolites while others do not. How could this arise and what are the implications? In general, what are the implications of a metabolite having a strong shadow price for an important reaction? I was surprised to see that arginine and ornithine do not come up in the shadow price analysis, as I would naively think that they would limit the transport reactions. The authors could expand on this in the results section here or in the discussion.

9. Line 145. The blue column seems to be a fraction of models not the number of models.

10. Line 143. Figure 3b. It would be good to include the ID of the metabolites or some other more specific name.

11. Paragraph 155-177. The MTA sections seems to be only weakly connected to the previous results from the ML section. Any efforts to link the two results in the discussion would be appreciated. (See major comment 4)

12. Line 183. Include the value of epsilon in the caption.

13. Paragraph 240-250. Good discussion of implications of the RIPTiDe algorithm. Additional discussion should be added regarding the limitations of GENRES and the other analyses utilized here.

14. Line 455. Sup Fig S2. An additional PCA plot with high and low toxin as the colors would be good to include here.

15. Lines 488-489. A numbering system is mentioned in the figure caption, but I do not see any use of that numbering system in the table. Maybe it would be clearer to include the RIPTiDe model reference in the table.

16. Line 348. The github repository that is linked seems to be empty at this time.

Reviewer #2: This is a review of “Network analysis of toxin production in Clostridioides difficile identifies key metabolic dependencies” by Powers and colleagues. This paper focuses on Clostridium difficile, a notorious opportunistic pathogen which has a diverse metabolism that enables it to establish a niche within the complex gut environment. Its pathogenesis is primarily mediated by toxins TcdA and TcdB. This paper successfully demonstrates how toxic production is regulated by the organism's metabolism, a long-standing question in the field. The study employs a system biology-based workflow, utilizing genome-scale metabolic models, to reveal how different extracellular environments can affect the regulation of toxin production and how this relates to changes in intracellular metabolism.

This is a strong paper especially because of its innovative use of publicly available transcriptomic data from various studies to provide an extracellular context for the genome-scale metabolic models. The choice of low and high toxin states through transcriptomic data gives these states the context while performing metabolic modelling using genome scale metabolic models. The contextualization of these models has shown that how these states are influenced by both extracellular and intracellular environments.

The use of the RIPTiDe algorithm and machine learning methods highlights the key role that arginine and ornithine, which are available from the environment, play in regulating toxin production. The paper goes on to show which is further regulated by intracellular pools of fatty acids and large polymer metabolite pools, as shown by the flux balance analysis and shadow pricing analysis. Additionally, the application of the mMTA algorithm identifies important reactions involved in transitioning from high to low toxin production states, providing ideas for potential therapeutic targets.

I am convinced that this paper will be a valuable contribution to the field. However, there are important issues that should be addressed before publication:

Major issues:

1. The paper states that the analysis code is shared but the associated link is empty. This needs to be fixed before publication.

2. The methods section could be more detailed in describing the usage and caveats of the RIPTiDE algorithm and shadow pricing analysis with respect to flux sampling to improve the reproducibility of the work.

Minor issues:

3. The paper is a valuable contribution to the field's understanding of toxin regulation by metabolism in C. difficile. However, it would have been beneficial to include a discussion of isoleucine fermentation as a source of energy metabolism in the results section.

4. The abstract and introduction effectively summarize the work and highlight its potential therapeutic applications, but the results and discussion sections could benefit from more specific recommendations on this subject.

5. Additionally, the figure legends in Figure 3A could be clearer, and the use of the term "context dependent" for metabolites with shadow price>2 may be confusing.

6. The legend in Figure 4 should also be more descriptive to make it accessible to non-experts in computational methods, and the use of the term epsilon without further explanation may be confusing to readers.

Reviewer #3: This paper report integration of publicly available transcriptomic data using RIPTiDe algorithm to create contextualize how nutritional changes regulate toxin levels in C. difficile. This work is built on previously published genome scale models of strains 630 and and R20291. While it is interesting to see transcriptomic data integration into the previously published model, it is incremental work. The following aspects could make this paper much stronger.

1. Integrate actual toxin level data following growth in Biolog Phenotype array previously reported by Lei and Bochner 2013. That paper is cited as ref#26 in the discussion. However, that data is not integrated with the model. There is some agreement between the results in the current work and phenotypic toxin data published in ref#26. For example, Arg-dipeptides showed higher toxin production in Biolog plates. Calibrating the FBA model with validated phenotypic toxin data will make this work much better. Since authors have done that previously in E. coli (data guided FBA model), doing that here should not be that difficult.

2. The other question is how genome variation in C. difficile affects toxin production. CD 630 and CD R20291 belong to different toxinotypes. There have been conflicting reports including and excluding the importanc of toxinotypes in C. difficle virulence. Although there is high genome variation in C. difficile, genes coding for the central metabolic pathways are somewhat conserved, albeit with some sequence variation. Is data integration with genome scale modeling sensitive enough to understand how the pathogenicity locus (where cluster of toxin genes are located) interacts with master regulators of central metabolic pathways (codY, CcpA, and others). Integrating Biolog data could be interesting because adenine and related compounds were the strongest toxin inducers in the previous study.

3. Public data set is available where both transcriptome and metabolome were analyzed following C. difficile infection in a mouse model (PMID: 29600278). Unfortunately, toxin levels were not reported in that paper. However, toxin gene expression level could be pulled from the transcriptome data, and do the predictions here correlate with what is reported in that work? I agree that dataset may not amenable for modelling, but could be useful in expanding the discussion in the context of toxinotype variation.

**Have the authors made all data and (if applicable) computational code underlying the findings in their manuscript fully available?**

Reviewer #1: **No: **The github repository is currently empty. The authors will need to upload their models and code here before the paper is published.

Reviewer #2: **No: **The links for the code and the data are made available in the paper but those links are empty and nothing is uploaded on them so far.

Reviewer #3: Yes

PLOS authors have the option to publish the peer review history of their article (what does this mean?). If published, this will include your full peer review and any attached files.

Reviewer #1: No

Reviewer #2: **Yes: **Vishwas Mishra

Reviewer #3: No
---

## [Decision Letter · Decision Letter 1]

4 Apr 2023

Dear Professor Papin,

We are pleased to inform you that your manuscript 'Network analysis of toxin production in Clostridioides difficile identifies key metabolic dependencies' has been provisionally accepted for publication in PLOS Computational Biology.

Best regards,

Kiran Patil

Section Editor

PLOS Computational Biology

Reviewer's Responses to Questions

**Comments to the Authors:**

Reviewer #1: The authors have done a good job of updating their manuscript following the first round of reviews. In particular, I am happy with the updates to the machine learning methods and discussion. The github is also now available with the code. I encourage the authors to include some additional documentation in a github readme file, but that should not hold up publication.

Reviewer #2: The authors have very well taken into account the reviewer's comments and have made necessary changes to the previous manuscript submission. This paper successfully shows how toxic production is regulated by the organism's metabolism. This is a great paper to be added to the field because of its novel way to make use of publicly available transcriptomic data to provide an extracellular context for the genome-scale metabolic models. I am convinced that this paper will be a valuable contribution to the field.

Reviewer #3: Authors have incorporated most of the suggestions made earlier. I don't have any further comments

**Have the authors made all data and (if applicable) computational code underlying the findings in their manuscript fully available?**

Reviewer #1: Yes

Reviewer #2: Yes

Reviewer #3: Yes

PLOS authors have the option to publish the peer review history of their article (what does this mean?). If published, this will include your full peer review and any attached files.

Reviewer #1: No

Reviewer #2: No

Reviewer #3: No

---

## [Editor Report · Acceptance letter]

20 Apr 2023

PCOMPBIOL-D-22-01878R1 

Network analysis of toxin production in Clostridioides difficile identifies key metabolic dependencies

Dear Dr Papin,

I am pleased to inform you that your manuscript has been formally accepted for publication in PLOS Computational Biology. Your manuscript is now with our production department and you will be notified of the publication date in due course.

With kind regards,

Zsofi Zombor
